# CFTR Function Restoration upon Elexacaftor/Tezacaftor/Ivacaftor Treatment in Patient-Derived Intestinal Organoids with Rare *CFTR* Genotypes

**DOI:** 10.3390/ijms241914539

**Published:** 2023-09-26

**Authors:** Juliet W. Lefferts, Marlou C. Bierlaagh, Suzanne Kroes, Natascha D. A. Nieuwenhuijze, Heleen N. Sonneveld van Kooten, Paul J. Niemöller, Tibo F. Verburg, Hettie M. Janssens, Danya Muilwijk, Sam F. B. van Beuningen, Cornelis K. van der Ent, Jeffrey M. Beekman

**Affiliations:** 1Department of Pediatric Respiratory Medicine, Wilhelmina Children’s Hospital, University Medical Center, Utrecht University, 3584 EA Utrecht, The Netherlands; 2Regenerative Medicine Utrecht, University Medical Center, Utrecht University, 3584 CT Utrecht, The Netherlands; 3Centre for Living Technologies, Alliance TU/e, WUR, UU, UMC Utrecht, 3584 CB Utrecht, The Netherlands; 4Department of Pediatrics, Division of Respiratory Medicine and Allergology, Erasmus Medical Center-Sophia Children’s Hospital, University Hospital Rotterdam, 3015 CN Rotterdam, The Netherlands

**Keywords:** cystic fibrosis, CFTR modulator therapy, intestinal organoids, elexacaftor/tezacaftor/ivacaftor, theratyping, theranostics, rare genotypes

## Abstract

Cystic fibrosis (CF) is caused by mutations in the *Cystic Fibrosis Transmembrane conductance Regulator* (*CFTR*) gene. The combination of the CFTR modulators elexacaftor, tezacaftor, and ivacaftor (ETI) enables the effective rescue of CFTR function in people with the most prevalent F508del mutation. However, the functional restoration of rare *CFTR* variants remains unclear. Here, we use patient-derived intestinal organoids (PDIOs) to identify rare *CFTR* variants and potentially individuals with CF that might benefit from ETI. First, steady-state lumen area (SLA) measurements were taken to assess CFTR function and compare it to the level observed in healthy controls. Secondly, the forskolin-induced swelling (FIS) assay was performed to measure CFTR rescue within a lower function range, and to further compare it to ETI-mediated CFTR rescue in CFTR genotypes that have received market approval. ETI responses in 30 PDIOs harboring the F508del mutation served as reference for ETI responses of 22 PDIOs with genotypes that are not currently eligible for CFTR modulator treatment, following European Medicine Agency (EMA) and/or U.S. Food and Drug Administration (FDA) regulations. Our data expand previous datasets showing a correlation between in vitro CFTR rescue in organoids and corresponding in vivo ppFEV1 improvement upon a CFTR modulator treatment in published clinical trials, and suggests that the majority of individuals with rare *CFTR* variants could benefit from ETI. CFTR restoration was further confirmed on protein levels using Western blot. Our data support that CFTR function measurements in PDIOs with rare *CFTR* genotypes can help to select potential responders to ETI, and suggest that regulatory authorities need to consider providing access to treatment based on the principle of equality for people with CF who do not have access to treatment.

## 1. Introduction

Cystic fibrosis (CF) is an autosomal recessive disorder caused by mutations in the *Cystic Fibrosis Transmembrane conductance Regulator* (*CFTR*) gene, which encodes for the CFTR protein [1,2]. Aberrant CFTR protein function limits luminal chloride and bicarbonate secretion, leading to altered epithelial fluid transport characteristics and ultimately multiorgan pathology. At present, over 2100 different *CFTR* variants have been documented, of which over 700 are known to be CF-causing [3]. The F508del mutation, a deletion of the amino acid phenylalanine at position 508, is the most common *CFTR* mutation. It is estimated that 80% of people with CF (pwCF) carry at least one F508del copy; all but five other mutations have (ultra-) low allele frequencies under 1% [3].

Over the past decade, CFTR modulating therapies have significantly transformed the treatment options for pwCF [4,5,6]. These drugs are small molecules that directly target the aberrant CFTR protein, leading to improved CFTR folding and subsequently increased CFTR trafficking (correctors) and channel opening (potentiators), thereby restoring its function [7,8,9,10]. Modulator therapies include the potentiator ivacaftor for the treatment of gating mutations and selected residual function mutations [11,12], whereas the combination of corrector and potentiator lumacaftor/ivacaftor (LUM/IVA) and tezacaftor/ivacaftor (TEZ/IVA) are available for pwCF homozygous for the F508del mutation [13,14]. Lastly, the highly effective triple combination elexacaftor/tezacaftor/ivacaftor (ELX/TEZ/IVA or ETI) has been approved for pwCF who carry at least one F508del mutation by the European Medicines Agency (EMA) [15] or either an F508del mutation or one of the 177 listed rare mutations by the United States Food and Drug Administration (FDA) [16]. These 177 rare mutations have not been selected from clinical studies, but from functional studies in Fisher Rat Thyroid (FRT) cells that show CFTR restoration beyond the 10% wild-type (WT) CFTR function threshold [17,18].

An unmet need remains to identify all individuals who might benefit from market-approved CFTR modulator therapy. The CF community is therefore also exploring how functional studies of CFTR modulators in patient-derived cells can be used as a ‘theranostic’ tool, to predict the likelihood of treatment benefit for individuals based on treatment response in their cultured cells [19,20,21]. This approach enables the measurement of the CFTR modulator response of both *CFTR* variants within the individual genetic background. This approach can complement the more reductionist approach provided by the FRT system, in which mutations are identified as treatment responsive (theratyping) and individuals are selected based on their CFTR DNA sequence analysis. 

Patient-derived intestinal organoids (PDIOs) have emerged as a relevant in vitro model to study baseline CFTR function and response to CFTR modulators, both at the mutation level and individual level [22,23,24]. PDIOs from pwCF are distinct from healthy donor intestinal organoids by CFTR-dependent luminal fluid secretion phenotypes that can be observed under standard culture conditions or upon stimulation of PDIOs with forskolin that opens the CFTR ion channel [23,25,26]. Fluid secretion assays such as steady-state lumen area (SLA) and rectal organoid morphology analysis (ROMA) measurements allow for the quantification of CFTR function at levels that differentiate between healthy- and CF-intestinal organoids [25,26]. Forskolin-induced swelling (FIS) can be used to quantify CFTR function and the response to modulators at a lower CFTR function level compared to SLA, and covers CFTR function ranging from severe CF to borderline CF [23]. Variation in CFTR function by FIS has been shown to associate with long-term disease progression and can have prognostic values for pwCF with unclassified *CFTR* variants [27]. Additionally, the association between FIS and the short-term clinical response to CFTR modulators has been shown at the group and individual level, but not when drug responses of PDIOs are compared to short-term clinical outcomes of individuals who carry identical mutations that respond to CFTR modulators at a population level [25,28,29,30].

In this study, we perform CFTR function measurements with ETI in PDIOs derived from pwCF harboring mutations that are not currently eligible for CFTR modulator treatment under FDA or EMA regulations. We compared CFTR restoration in PDIOs with rare genotypes to healthy control rectal organoids or in response to ETI in F508del-PDIOs. We conclude that discrepancies exist between PDIO and FRT results and FDA approval for CFTR modulator therapy. Using PDIOs, we identified additional rare variants responsive to market-approved CFTR modulators.

## 2. Results

### 2.1. Experimental Approach for the Identification of ETI-Responsive CFTR Genotypes in PDIOs

We set out to identify rare *CFTR* mutations that show an increase in CFTR function upon ETI modulator treatment (experimental approach outlined in Figure 1). Organoids were pre-incubated with the correctors ELX/TEZ, and 24 h later forskolin (dose range) and ivacaftor (IVA) were added acutely to measure forskolin-induced swelling (FIS) within an hour. First the steady-state lumen area (SLA) phenotype was analyzed on t = 0 of the FIS assay for both the ETI and DMSO (vehicle; negative control) condition. These effects of a corrector pre-treatment on PDIOs were compared to SLA phenotypes of healthy control organoids. Secondly, FIS measurements in PDIOs were analyzed and compared to functional CFTR restoration of F508del/class I in response to ETI treatment. 

### 2.2. CFTR Modulator Effects in Rare Variant CF PDIOs in Relation to Wild-Type CFTR Function 

First, we studied the effect of ELX/TEZ incubation on luminal fluid secretion in PDIOs with rare *CFTR* genotypes to quantify corrector-mediated function restoration after overnight incubation prior to the FIS assay. All vehicle-treated PDIOs had small organoid lumens, consistent with limited CFTR function. Overnight incubation with ELX/TEZ increased organoid swelling, in a donor-dependent manner, prior to forskolin and ivacaftor treatment (Figure 2A, FIS t = 0). We quantified this swelling with SLA; this revealed that the luminal organoid area was below 10% of the total organoid area for all vehicle-treated PDIOs (Figure 2B) [25]. Upon overnight corrector incubation, F508del/class I (*n* = 3) and F508del/F508del (*n* = 3) PDIOs and most of the PDIOs with rare *CFTR* genotypes showed SLA of <40%, indicating functional rescue below WT levels (Figure 2B). However, PDIOs with the genotypes R1066C/R1066H, R1066H/CFTRdele2,3, and Q1012P/N1303K showed SLA values of 47, 56, and 58%, respectively, within the range of historical data of healthy control donors [25,31]. Additionally, Western blot analysis confirmed an increase in mature CFTR C-band protein upon ETI treatment in these donors (Appendix A). However, we did not observe a correlation between relative CFTR C-band protein and SLA levels for all donors (Appendix A), potentially caused by *CFTR* variants that require further potentiating for restoration of CFTR function. SLA increased in the majority of the donors upon overnight ELX/TEZ incubation; however, only three donors reached SLA levels comparable to healthy control organoids.

### 2.3. In Vitro CFTR Rescue by ETI Modulator Therapy Correlates with In Vivo Change in ppFEV1 

The FIS assay was used to quantify the CFTR modulator response at lower CFTR-function levels. We first established reference values for functional CFTR restoration by ETI in PDIOs harboring the F508del mutation. The in vitro rescue of CFTR function was assessed using the FIS assay at four different forskolin concentrations in F508del/class I (*n* = 15, for a complete overview of class I genotypes, see Appendix A) and F508del/F508del (*n* = 15) PDIOs (Figure 3A). The FIS response to ETI therapy was significantly increased in all individual donors at all forskolin concentrations. Swelling at 0.128 µM forskolin allowed for the best differentiation between FIS responses and has previously been shown to correlate with in vivo parameters [25,28]. Mean swelling at 0.128 µM forskolin was significantly higher (*p* < 0.05) in F508del/F508del PDIOs (2433 ± 525; mean ± SD AUC) than in F508del/class I (1957 ± 654; mean ± SD AUC) (Figure 3B). Western blots further confirmed an increase of mature CFTR protein expression upon CFTR modulator treatment. We measured immature core-glycosylated B-band (CFTR-B) and mature complex glycosylated C-band (CFTR-C) CFTR for ETI- and DMSO-treated F508del/class I and F508del/F508del PDIOs (Figure 3C). CFTR C-band levels, relative to the total CFTR levels (CFTR-B + CFTR-C), were low for untreated conditions and increased upon ETI treatment in both genotypes (Figure 3D). To interpret the relation between in vitro FIS response and clinical benefit, DMSO-corrected FIS responses of F508del/class I and F508del/F508del PDIOs were correlated to absolute change in percentage of predicted forced expiratory volume in 1 second (ppFEV1) versus placebo in vivo, from currently available clinical trial data [12,13,32,33,34,35,36,37], and added to historical in vitro and in vivo data of other modulator treatments in various *CFTR* genotypes (Figure 3E). We observed a correlation between in vitro and in vivo CFTR rescue, indicating that the amplitude of FIS response is associated with clinical response, on a group level. Together, the data support the use of FIS response at 0.128 µM forskolin in F508del PDIOs as a reference for response-to-ETI in PDIOs with rare *CFTR* variants. 

### 2.4. CFTR Modulator Effects in Rare Variant CF PDIOs in Relation to F508del ETI Induced Function

Finally, we performed measurements to assess the functional and molecular response in PDIOs with rare *CFTR* variants that had SLA levels below healthy control levels after overnight ELX/TEZ incubation. FIS was performed in the presence of DMSO or ETI to measure the CFTR baseline function and response-to-therapy, respectively, at four different forskolin concentrations. FIS at 0.128 µM forskolin increased upon ETI treatment, compared to the vehicle treatment, in 15 out of 19 PDIOs (Figure 4A; overview of all FIS data per individual line in Appendix A). The modulator response was within the range of the mean F508del/class I response-to-ETI minus 1 SD (1957–654; mean −1 SD AUC) for nine of these genotypes. An additional four PDIOs showed swelling comparable to the mean F508del/F508del response to LUM/IVA minus 1 SD (1230–419; mean −1 SD AUC), based on historical data. Table 1 provides a summary of responses and compared them to both published FRT responses [38] and the FDA label extension. Western blots were performed to assess the relative CFTR C-band levels. Some donors (e.g., [R334W;Q378X]/[R334W;Q378X]) showed profound C-band levels in the absence of ETI with low-to-absent residual CFTR function measured with FIS. A relation was shown between relative CFTR-C protein levels upon ETI treatment and the FIS response to ETI (Figure 4B,C; an overview of all Western blots in Appendix A). These data indicate that ETI can potentially restore CFTR function in 13 out of 19 donors towards magnitudes associated with or beyond CFTR function levels of F508del PDIOs treated with LUM/IVA.

## 3. Discussion

In this study, we used PDIOs of people who are not eligible for treatment and studied how their functional restoration by ETI compared to in vitro WT CFTR function and ETI responses in FDA and EMA market-approved CFTR genotypes. Our data show that 16 out of 22 PDIOs show a considerable-to-high response to ETI, of which 3 donors showed CFTR function increase upon overnight ELX/TEZ incubation compared to WT PDIOs, and 13 PDIOs showed CFTR function restoration in response to ETI treatment comparable to or higher than the F508del/F508del response to LUM/IVA in the FIS assay. This supports the potential clinical benefit of ETI for a substantial fraction of pwCF who are not eligible for modulator treatment. 

We have used two CFTR-dependent readouts to analyse the functional response to ETI in rare *CFTR* genotypes. SLA has a dynamic range at higher CFTR function levels than FIS, and discriminates CF organoids from healthy controls [25]. This facilitates the comparison of the CFTR modulator response to healthy control CFTR function. FIS quantifies CFTR function at CFTR function levels associated with severe CF towards ‘borderline’ CF, as demonstrated by the strong association between FIS and annual pulmonary function decline and the odds to develop CF-specific comorbidities [27]. The combination of assay formats allows for CFTR function measurements with a large dynamic window and for the comparison of CFTR function (restoration) to healthy controls and clinically-relevant CFTR genotype-drug combinations. 

Three out of twenty-two PDIOs with rare *CFTR* mutations had SLA levels of >51% upon overnight ELX/TEZ incubation, indicative of CFTR function within the range of WT CFTR function. A further 12 PDIOs showed considerable levels of endogenous swelling (10–35% SLA), beyond the SLA levels associated with a CF diagnosis [25]. Consistent with the CF diagnosis, PDIOs with moderate-to-high levels of ELX/TEZ-induced SLA all had baseline SLA levels < 10% in the presence of vehicle treatment and harbored *CFTR* variants that have been assigned as CF-causing except for the Q1012P variant, which is not listed in the CFTR2 database [3]. Additionally, these variants are predominantly severe class II trafficking mutations, except for the 4382delA mutation which has been associated with residual CFTR function [27]. Genotypes with an SLA of <10% upon ELX/TEZ treatment include nonsense, severe splice, and/or class II variants. Taken together, this suggests that ELX/TEZ-induced CFTR function increase can be measured by the SLA of organoids, can enable a comparison to WT CFTR function, and suggests that 3 out of 22 PDIOs may benefit strongly from ETI treatment. 

A previous study demonstrated a negative association between high SLA (>40%) and organoid swelling [25]. Indeed, the FIS response in the PDIOs with high SLA upon ELX/TEZ incubation was low and not representative of the functional CFTR restoration (Appendix A). PDIOs with SLA values of >40% upon ELX/TEZ incubation were therefore excluded from the FIS analysis. In our study, we specifically chose to study functional responses by SLA upon overnight ELX/TEZ incubation, in the absence of IVA, and added IVA together with forskolin as conducted in previous works and other model systems (e.g., organoids or FRT model systems). The overnight addition of IVA has previously been shown to further enhance the endogenous organoid expansion, which may reduce PDIO numbers that can be measured by FIS due to the negative impact of SLA on FIS at a high SLA [25]. We currently prefer FIS as this assay is better validated than SLA, easier to execute, and more suited to quantify CFTR function at a level where clinical benefit can start to be expected (between the severe and borderline CF range, or CFTR function associated with LUM/IVA in F508del homozygosity). This current experimental approach may not fully appreciate the long-term effects of IVA on the CFTR protein that can differentially impact on rare CFTR mutations, and has also been reported to destabilize the corrector-rescued CFTR protein [39]. Additional image analysis strategies for SLA may help to develop a fully automated, unbiased assay platform to measure SLA and FIS under chronic ETI conditions to compensate for such effects.

Functional CFTR restoration in PDIOs with SLA levels of <40% upon 24 h corrector incubation was further assessed using the FIS assay. We first measured the FIS response to ETI treatment in F508del/class I (*n* = 15 donors) and F508del/F508del (*n* = 15 donors) PDIOs, to use as a benchmark for clinically significant CFTR restoration and the definition of CFTR function response associated with current market-approved conditions. All individual donors showed a significant response to ETI treatment [32,35]. Responses were at the higher end of the dynamic range of the FIS assay, as there was a plateau in swelling. A significant difference between the F508del/class I and F508del/F508del responses was observed at lower forskolin concentrations (0.128 µM), but the magnitude of the functional difference between these genotype groups was lower as previously observed for LUM/IVA [25]. The FIS response at 0.128 µM forskolin of various modulator conditions and paired representative clinical data from the literature showed a clear correlation, as also identified previously and by others [25,28]. It must be noted that the in vivo ETI response in the F508del homozygous group is probably underestimated due to the TEZ/IVA therapy background [35]. These data support that FIS at 0.128 µM forskolin measures relevant ETI-mediated CFTR function restoration in F508del-PDIOs, facilitating the comparison to CFTR function restoration in PDIOs with rare variants. 

Next, we assessed the swelling response in the PDIOs with rare genotypes at 0.128 µM forskolin. Nine PDIOs reached FIS levels comparable to, or higher than the mean F508del/class I response to ETI; another four PDIOs showed an FIS response in the range of the F508del/F508del response to LUM/IVA, suggesting potential for clinical benefit. PDIOs with moderate SLA levels were all identified as high FIS responders, except for 2043delG/4382delA PDIOs, which had a moderate FIS response, within the F508del/F508del LUM/IVA range. All PDIOs that did not show a response in the FIS assay were also non-responders upon overnight ELX/TEZ incubation, as measured with SLA. Combining the SLA and FIS data, we identified 16 PDIOs containing 14 unique genotypes, as prospective CFTR modulator therapy responders. 

PDIO responses were compared to recent published FRT data where 655 rare CFTR variants were tested to ETI [38] and FDA regulations (Table 1). Previous studies have shown a relation between fluid secretion and ion transport studies in PDIOs [23,40]. However, ion transport measurements in 2D cultures have a higher dynamic range than the FIS assay [41]; we therefore employ the combined approach of FIS and SLA measurements in this study. In the future, ion transport studies in 2D cultures of these patient-derived intestinal cultures could be performed to determine if the discrepancies between PDIO and FRT results are caused by genetic of technical differences. All genotypes, present in this study, that have been classified as eligible for CFTR modulator therapy by the FDA have modulator response within the range of the F508del/class I response to ETI, except for the L1335P/L1335P genotype. L1335P homozygous PDIOs showed a low response to ETI, consistent with results in the FRT model [38]. More importantly, we have detected significant CFTR restoration in PDIOs harboring genotypes that have not been indicated as eligible for CFTR modulator therapy by the FDA (and EMA). These include PDIOs homozygous for the [R334W;Q378X] complex allele, compound heterozygous for the Q1012P/N1303K mutations, and compound heterozygous for the R334W mutation, which has recently also been identified as responsive to ETI [42] and showed great clinical response (ppFEV1 +38% and sweat chloride concentration (SwCl) −26 mmol/L) in an individual described in a case study [43]. Additionally, we found a significant response to ETI in PDIOs harboring the G85E variant, whereas the G85E was unresponsive to ETI treatment in FRT studies [38]. In contrast, PDIOs harboring the L927P variant were unresponsive to ETI treatment, while this variant has been identified as ETI-responsive in FRT studies. The discrepancy between these results could potentially be caused by an intronic *cis*-acting variant, which is not recapitulated in the FRT model, as the L927P variant has been associated with a variant in IVS7 [44]. Finally, donors homozygous for the N1303K mutation showed an ETI response below the F508del/class I ETI threshold but within the range of previously measured LUM/IVA responses in F508del/F508del PDIOs. This is in line with previously published results where ETI shows great improvement clinically, mainly improvement in ppFEV1 but little-to-no improvement in SwCl and a moderate response in PDIOs [45,46]. This demonstrates that further exploration of the in vivo ETI response in people with the N1303K mutation could be very valuable, especially since this mutation has an allele frequency of 1.6% [3].

Here, we have used group-level responses in F508del/class I and F508del/F508del to ETI and LUM/IVA (mean −1 SD), respectively, as an indicator of functional CFTR restoration in the FIS assay. This method for defining organoid thresholds could be used for the label extension of CFTR-modulators to people who are not eligible, based on their equality to CFTR function in organoids from people for who these drugs are available. This approach does not enable the definition of the test sensitivity and specificity for identifying individual clinical responders, which remains a highly challenging approach due to the lack of clinical data in true negative responder populations and the difficulty to accurately establish individual clinical benefit in the short term. A lower threshold for response could be, as we have piloted in this study, −1 SD for the group of pwCF who are eligible for CFTR modulator therapy and who have the lowest clinical benefit (for ETI:F508del/class I; for LUM/IVA: F508del/F508del), but additional statistical strategies or thresholds might be considered. 

Taken together, we conclude the discrepancies between PDIO and FRT results and FDA approval for CFTR modulator therapy. We propose a combined approach of SLA and FIS to measure in vitro CFTR function restoration in response to ETI. We show CFTR function restoration in 12 PDIOs with 11 unique genotypes by ETI therapy, compared to WT CFTR function or to the F508del/class I ETI response. We have detected functional CFTR restoration in genotypes harboring the A46D, G85E, E92K, R1066H, and S1159F variants, which have been approved for modulator therapy by the FDA, but not by the EMA. Additionally, we also identify the R334W, Q1012P, N1303K, and other unique combinations of variants within functional responsive genotypes, which have not been approved for modulator therapy by the FDA and the EMA. The relation between the in vitro PDIO response and in vivo parameters, although on a group level, suggests that these pwCF may benefit from ETI modulator treatment in vivo. These data support the use of PDIOs to identify potential responders to CFTR modulators. We also urge authorities to explore the path towards drug access for people with rare *CFTR* variants on the principle of equality to pwCF who do have access to drugs.

## 4. Material and Methods

### 4.1. Ethical Approval for Organoid Use

The organoids used were obtained from the Foundation Hubrecht Organoid Biobank (Utrecht, The Netherlands) under TC-Bio protocol number 14-008, or from the UMCU Darmbank under TC-Bio protocol number 19-831, and used according to informed consent. 

### 4.2. PDIO Selection Rare Genotypes

PDIOs were selected from the HUB biobank. A total of 15 F508del/F508del, 15 F508del/class I (nonsense), and 22 rare PDIOs were selected. The selection criteria for rare PDIOs were as follows: PDIOs could not harbor any *CFTR* variant eligible for CFTR modulator treatment, according to EMA regulations, at the time of PDIO selection (Dec 2021). Additionally, PDIOs could not harbor two minimal function *CFTR* variants. The criteria for minimal function variants were: (i) a premature stop codon before p.1282, (ii) a splice site mutation +1/2 or −1/2 from the intron/exon junction, (iii) a frameshift mutation (insertion/deletion) below c.3846, or (iv) a full exon deletion. This resulted in a PDIO selection of 22 samples.

### 4.3. PDIO Culture

PDIO culture was performed as described before [47]. In short, organoids were maintained in 40% matrigel droplets in the presence of WNT-conditioned culturing medium. Culturing medium was refreshed three times per week, and organoids were passaged weekly, by manual disruption, cleaning, and reseeding. 

### 4.4. Steady-State Lumen Area Quantification

To determine the steady-state lumen area (SLA), organoid images of the first data point of FIS measurements (FIS t = 0) were used. Both the organoid and luminal area were quantified per well in Labelbox, by the manual labelling and reviewing of organoid and lumen areas by organoid experts [48]. SLA was expressed as the percentage luminal organoid area of the total organoid area per well. 

### 4.5. Forskolin-Induced Swelling 

The CFTR baseline function and response-to-therapy were measured using the forskolin-induced swelling (FIS) assay. Organoids were cultured for at least three weeks before performing the first FIS measurement. Each PDIO was measured at three independent time points using technical duplicates per measurement. FIS measurements were performed as described previously [47]. In short, organoids were seeded into 96-well plates and treated with CFTR corrector compounds, ELX (MedChemExpress, Monmouth Junction, NJ, USA), and TEZ (SelleckChem, Houston, TX, USA) overnight. The following day, organoids were stained using calcein green (0.84 µM), before adding potentiator, IVA (SelleckChem, Houston, TX, USA) compounds, and forskolin to activate CFTR. All CFTR modulators were used at a final concentration of 3 µM and DMSO (vehicle) was added to all conditions to normalize to the highest concentration. Organoid swelling was imaged for an hour with 10 min intervals using a Zeiss LSM800 confocal microscope. We used Zeiss Zen Blue imaging software (version 2.3) for the quantification of the total organoid size per well at each time point, and finally calculated the area under the curve (AUC) of organoid swelling as a measure of CFTR function. 

### 4.6. Western Blot

Western blots were obtained to detect CFTR protein. PDIO cultures were treated with ETI triple therapy (3 µM) or DMSO for 48 h. Per experimental conditions, three 24-wells were harvested and lysed in Laemmli buffer, supplemented with protease inhibitor. Protein concentration was quantified using a Pierce BCA protein assay kit, following manufacturers protocol. SDS-page was performed with a 50 µg protein per condition. Proteins were transferred from SDS-page gels to polyvinylidene difluoride membrane overnight. CFTR protein was detected with a mix of three mouse monoclonal antibodies with different epitopes, being 450, 570, and 596 (CFF). For secondary staining, rabbit-anti-mouse-HRP was used. Hsp90 was labelled as the loading control using goat-anti-rabbit-HRP as a secondary antibody. Images were acquired using a BioRad ChemiDoc Touch Imaging System (version 3.0.1). Blots were further processed and analyzed using ImageJ (version 1.53t). 

## Figures and Tables

**Figure 1 ijms-24-14539-f001:**
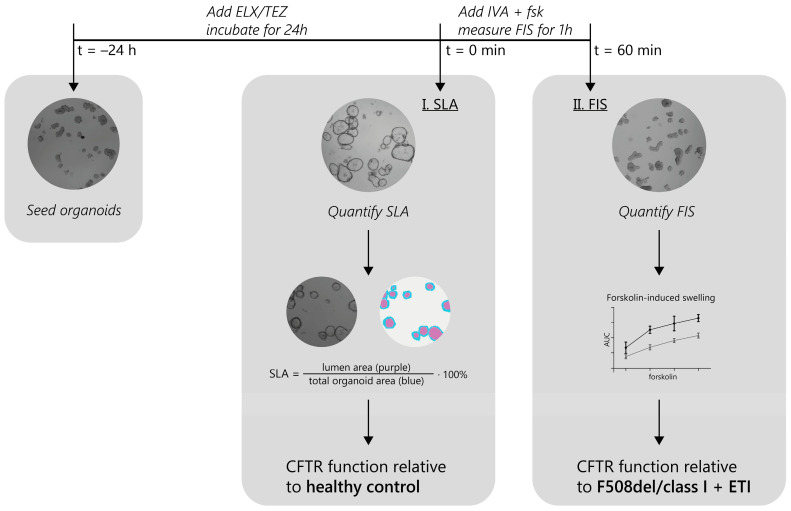
Schematic overview of CFTR function measurements in PDIOs.

**Figure 2 ijms-24-14539-f002:**
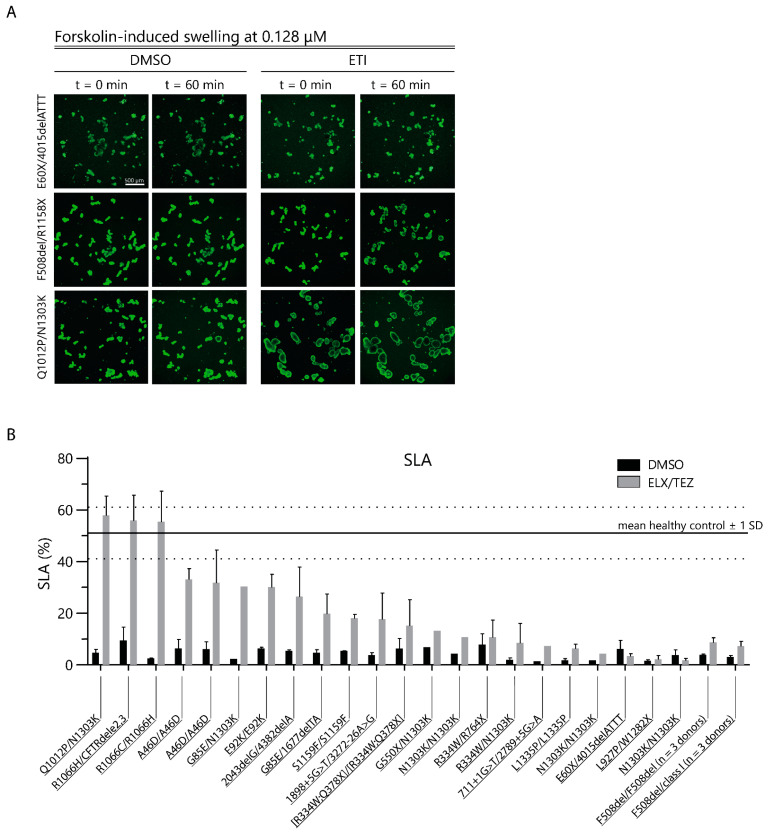
(**A**) Representative images of PDIOs at the start and end of FIS measurements in the presence of DMSO or ETI treatment at 0.128 µM forskolin. (**B**) Quantification of SLA upon overnight ELX/TEZ incubation, at t = 0 FIS assay, in three donors per F508del-genotype and PDIOs with rare *CFTR* variants. Healthy control range was used from historical data, dotted line represents +/−1 SD [25]. Error bars represent the SEM.

**Figure 3 ijms-24-14539-f003:**
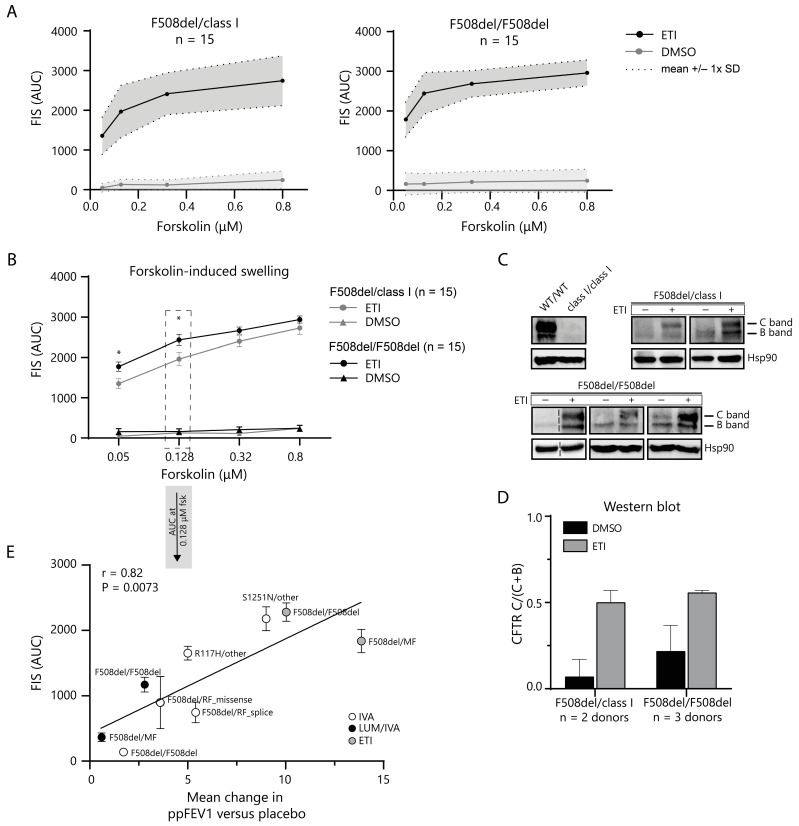
CFTR rescue in F508del/class I and F508del/F508del PDIOs in response to ETI treatment. (**A**) FIS performed in 15 donors per genotype which shows significant swelling for both genotypes at all forskolin concentrations in response to ETI treatment. (**B**) FIS response to ETI treatment in F508del/class I compared to F508del/F508del PDIOs. Error bars represent SEM, significance * = *p* < 0.05 and represent F508del/class I response compared to F508del/F508del response calculated using a two-tailed *t* test. (**C**) Western blots of F508del/class I and F508del/F508del PDIOs treated with ETI and positive (WT/WT) and negative (class I/class I) control intestinal organoids (**D**) Quantification of Western blots show an increase in relative levels of mature CFTR-C band for both F508del/class I and F508del/F508del PDIOs upon ETI treatment. Error bars represent the SD. (**E**) Pearson correlation of DMSO-corrected FIS response at 0.128 µM versus absolute change in ppFEV1, for multiple genotype-CFTR modulator combinations.

**Figure 4 ijms-24-14539-f004:**
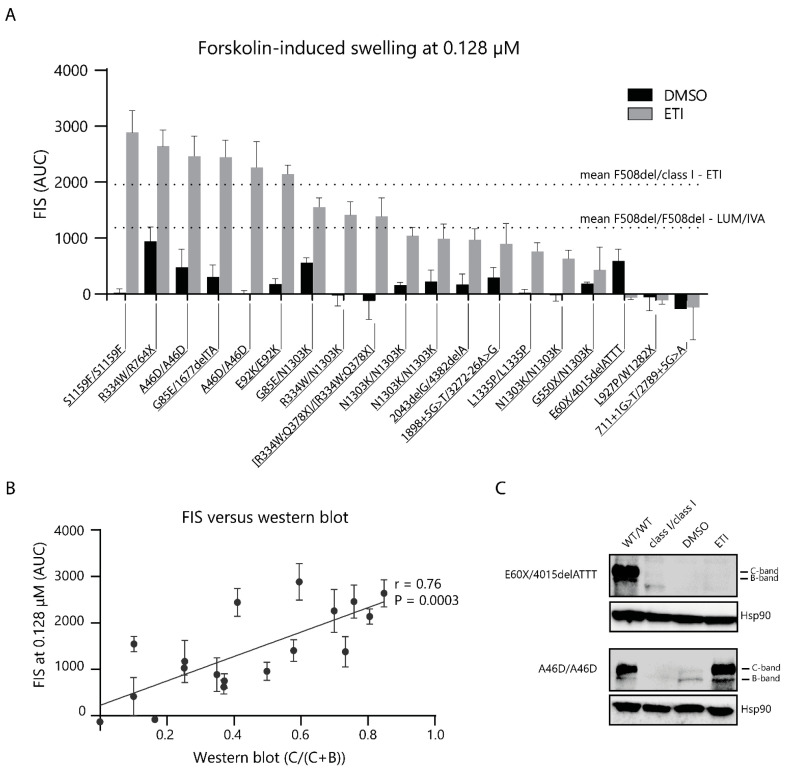
CFTR rescue upon ETI treatment in PDIOs harboring rare *CFTR* genotypes. (**A**) FIS at 0.128 µM forskolin compared to mean F508del/class I ETI and F508del/F508del LUM/IVA response. Error bars represent the SEM (n = 3). (**B**) Correlation between rescue of mature CFTR C-band and FIS at 0.128 µM forskolin for ETI-treated PDIOs. Error bars represent the SEM (n = 3). (**C**) Western blots of PDIOs non-responsive to ETI treatment, E60X/4015delATTT, and responsive to ETI treatment, A46D/A46D. Western blots include WT/WT intestinal organoids and class I/class I as positive and negative controls, respectively.

**Table 1 ijms-24-14539-t001:** PDIO responses and their mutations in relation to the recently published FRT responses or rare mutations to ETI and the FDA-label [16,38]. PDIO response is classified as: Healthy control (steady-state lumen area > 40%), ETI range (higher than mean F508del/class I response-to-ETI minus 1 SD, AUC 1303 at 0.128 µM forskolin), LUM/IVA range (higher than mean F508del/F508del response-to-LUM/IVA min 1 SD, AUC 811 at 0.128 µM forskolin), below LUM/IVA range (<811 AUC at 0.128 µM forskolin), and no swelling. X indicates if the *CFTR* variant on that allele is on the FDA-approved label of ETI.

Allele 1/Allele 2	PDIO Response	FRT ≥ 10% Allele 1	FRT ≥ 10% Allele 2	FDA-Approved Allele 1	FDA-Approved Allele 2
R1066C/R1066H	Healthy control	Non-responsive	Responsive		X
R1066H/CFTRdele2,3	Healthy control	Responsive	Not tested	X	
Q1012P/N1303K	Healthy control	Not tested	Borderline		
S1159F/S1159F	ETI range	Responsive	Responsive	X	X
R334W/R764X	ETI range	Non-responsive	Not tested		
A46D/A46D	ETI range	Responsive	Responsive	X	X
G85E/1677delTA	ETI range	Non-responsive	Not tested	X	
A46D/A46D	ETI range	Responsive	Responsive	X	X
E92K/E92K	ETI range	Responsive	Responsive	X	X
G85E/N1303K	ETI range	Non-responsive	Borderline	X	
R334W/N1303K	ETI range	Non-responsive	Borderline		
[R334W;Q378X]/ [R334W;Q378X]	ETI range	Not tested	Not tested		
N1303K/N1303K	LUM/IVA range	Borderline	Borderline		
N1303K/N1303K	LUM/IVA range	Borderline	Borderline		
2043delG/4382delA	LUM/IVA range	Not tested	Not tested		
1898 + 5G > T/3272-26A > G	LUM/IVA range	Not tested	Not tested		
L1335P/L1335P	Below LUM/IVA	Non-responsive	Non-responsive	X	X
N1303K/N1303K	Below LUM/IVA	Borderline	Borderline		
G550X/N1303K	No swelling	Not tested	Borderline		
E60X/4015delATTT	No swelling	Not tested	Not tested		
L927P/W1282X	No swelling	Responsive	Not tested		
711 + 1G > T/2789 + 5G > A	No swelling	Not tested	Not tested		

## Data Availability

The data presented in this study are available on request from the corresponding author.

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
