# Peer review of "CFTR Function Restoration upon Elexacaftor/Tezacaftor/Ivacaftor Treatment in Patient-Derived Intestinal Organoids with Rare CFTR Genotypes"

_ijms, 2023, doi:10.3390/ijms241914539_

Round 1

Reviewer 1 Report

This manuscript investigates PDIOs with CFTR mutations that are ineligible for current CFTR modulator therapies. Interestingly, a number of genotypes are shown by the authors to be responsive to triple-combination therapy (ETI).  This is of significance.

I feel the overall approach, using patient derived intestinal organoids is a suitable choice for this work.

In my view the results section could be improved with some additional detail and clarity concerning the experimental approach.  

Comments

Line 23: change Fist to First

Line 53: pwCF has already been defined (see line 49)

Line 114: should read “had small organoid lumens”; overall the sentence may be improved by simply saying something like the following….…All vehicle treated PDIOs had small organoid lumens, consistent with lack of CFTR function.

Line 116-117: states increased SLA in Fig 2A; however, Fig 2A seems to be a FIS assay. Is the text referring to the DMSO/t=0 readings? If so, this could be described more clearly.

Fig 1 & Fig 4: F508del/Class I + ETI is include as a benchmark. In Fig 4 F508del/F508del + Lum/Iva is also included. Is it possible to include F508del/F508del- ETI?

Line 117-118: I take it this refers to the DMSO data (<10%). Overall I think a more detailed description of Fig 2A and 2B in the results text would enhance clarity of this section. In Figure 2B it may be helpful to used label the y-axis in 10% increments.

Fig 2B: For the SLA measurements shown the CFTR corrector therapy comprises ET. For these assays why was Iva not included- considering the study focuses on ETI? As such, I am a bit confused by the figure and how to interpret in the scheme of the overall aim (identifying patients who may be suitable for ETI).

Note: From the discussion section it seems Iva was added acutely- this needs amended for clarity.

Line 125: The description of the blots in the results text is very brief and restricted to the 3 mutants that were restored by ET (Fig 2B). Further discussion may be of benefit- I note that…

·         R334W/R764X demonstrated no increase in SLA (%) upon ET treatment. Yet, the blot in S1 for this genotype shows a very nice increase in C-band levels with ETI.

·         Regarding R334W; Q378X there is quite a large C-band present prior to ETI which is elevated further with the CFTR modulator treatment. For this genotype SLA reading are quite modest.

Can the authors comment on the blots further?

Overall, I would say that it takes some time and effort to compare Fig 2A/Fig S1. This could perhaps be improved. Is there a reason not to include SLA vs WB similar to Fig 4B?

Line 137: is “here: nonsense mutations” meant to be included or necessary

Line 161: change PDIOS to PDIOs

Fig 4B: As above- can F508del/F508del-ETI be included as a benchmark

Line 177-179: mentions a range (and 9 genotypes identified in relation to the F508del/class I).  I see 6 genotypes above the marker for this mutation class so was unclear about this. Can the authors clarify this part of the text.

Line 206: Can you refer back to the results to clarify where 16 PDIOs have been shown as responsive to ETI?

Line 238-243: As far as I can see this is the first mention of the addition of Iva alongside Fsk when performing the SLA assay. I left my comment above (Fig 2) relating to this point. Can the authors ensure the experimental set up is clear in the results sections for both assays as I have found this hard to follow i.e. if and when Iva has been added?

Some minor improvements are required

Reviewer 2 Report

The manuscript by Lefferts and colleagues is an interesting study that presents new information about the discrepancies between PDIO and FRT results and FDA approval for CFTR modulator therapy, as well as extending the highly effective modulators therapy to patients who are deprived or do not have access to the modulators. Although the study is interesting and provides new information, some critical flaws must be addressed, as indicated in my specific comments.

Specific comments: 

1)    In fig 4A the effect of DMSO is nearly equivalent/touching to the LUMA/IVA mean F508del/F508del cutoff line. Does this imply that DMSO had a corrective effect as well? One could claim that the effect of R334W is attributable to a partial/residual function of the mutation.

2)    What does negative FIS values indicate? Is it only that there is no swelling? Or organoids are shrinking in size, resulting in a negative AUC. It would be ideal if the authors removed the negative values or just mention no swelling on the bar graph. 

3)    Scale bar from the FIS images is missing. 

4)    The previously published responses in FRT cells were obtained using Ion transport assay and complemented with protein quantification by Western blots. It would be interesting if the authors convert 3D organoids to 2D monolayer for Ion transport measurements (using Ussing chamber) in mutations that were responsive in FRT cells but irresponsive in organoids and vice versa, i.e. responsive in organoids but irresponsive in FRT cells. This will also rule out the possibility of different cells (FRT vs PDIO’s), or methodologies (Ussing chamber measurements vs FIS assay) being used to measure CFTR Cl- currents. 

5)    Effect of TRIKAFTA in non-CF organoids is not addressed in the manuscript. Although these compounds are tailored to CFTR mutations, the effect must be demonstrated in non-CF organoids as well. 

6)    Patient demographics is missing in the manuscript. 

Minor comments: 

1)    Check spelling on 1st page: line 23: “Fist” changes to “First”. 

2)    Figure 2A, mention the data represented in bar graph is SD or SEM in figure legend. 

3)    Check spelling on 12th Page: line 382 “rabibit” changes to “Rabbit”. 

4)    Line 300: “were” changes to “where”? 

5)    Line 98: “approve” changes to “approved”?

6)    It would be visually appealing if authors use color code for ETI and DMSO in figures. 

The manuscript is well written, and the text is clear and easy to read.

Round 2

Reviewer 2 Report

I have no further comments.